# DON'T BE GREEDY, JUST RELAX! PRUNING LLMS VIA FRANK-WOLFE

## ABSTRACT

*Pruning* is a common technique to reduce the compute and storage requirements of Neural Networks. While conventional approaches typically retrain the model to recover pruning-induced performance degradation, state-of-the-art Large Language Model (LLM) pruning methods operate layer-wise, minimizing the per-layer pruning error on a small calibration dataset to avoid full retraining, which is considered computationally prohibitive for LLMs. However, finding the optimal pruning mask is a hard combinatorial problem and solving it to optimality is intractable. Existing methods hence rely on greedy heuristics that ignore the weight interactions in the pruning objective. In this work, we instead consider the convex relaxation of these combinatorial constraints and solve the resulting problem using the Frank-Wolfe (FW) algorithm. Our method drastically reduces the per-layer pruning error, outperforms strong baselines on state-of-the-art GPT architectures, and remains memory-efficient. We provide theoretical justification by showing that, combined with the convergence guarantees of the FW algorithm, we obtain an approximate solution to the original combinatorial problem upon rounding the relaxed solution to integrality.

## 1 INTRODUCTION

*Pruning after training* (Han et al., 2015; Gale et al., 2019; Hoefler et al., 2021; Zimmer et al., 2023; 2025) reduces the inference-time compute and memory footprint of Neural Networks with minimal impact on predictive performance. Conventional approaches obtain such *sparse* models by removing parameters using simple criteria such as their magnitude and then typically require full retraining to recover pruning-induced performance degradation. The drastic increase in model size accompanying the rise of LLMs has, however, reshaped the pruning landscape.

At LLM scale, full retraining is often considered prohibitively expensive or even infeasible, resulting in a surge of interest in pruning criteria that do not require retraining. In addition, classical magnitude pruning performs no better than random pruning for LLMs (Sun et al., 2023; Yin et al., 2023), an observation attributed to activation outliers (Dettmers et al., 2022) and highly important *super-weights* (Yu et al., 2025) in sufficiently large *Transformer* models (Vaswani et al., 2017). Consequently, state-of-the-art methods (Frantar & Alistarh, 2023; Sun et al., 2023; Zhang et al., 2024) prune *layerwise*: they decompose pruning into per-layer subproblems and treat layers sequentially and independently, estimating parameter importance on a small calibration set by minimizing a per-layer *local* pruning loss. Specifically, for a single layer with calibration input matrix $X \in \mathbb{R}^{d_{in} \times B}$ and weights $W \in \mathbb{R}^{d_{out} \times d_{in}}$, the objective is

$$\min_{M} \|WX - (M \odot W)X\|_F^2, \quad \text{s.t. } M \in \{0, 1\}^{d_{out} \times d_{in}}, \|M\|_0 \leq k \quad \text{(MASK SELECTION)}$$

where $M \in \{0, 1\}^{d_{out} \times d_{in}}$ is a binary mask that enforces the target sparsity, e.g., $\|M\|_0 \leq k$ for unstructured pruning, and $\odot$ denotes the Hadamard product. Here, $B = N \cdot L$, where $N$ is the number of samples in the calibration batch and $L$ the sequence length.

However, even for a single layer, selecting the optimal pruning mask is a hard quadratic binary optimization problem. Solving (MASK SELECTION) to optimality is computationally intractable at LLM scale because the combinatorial constraint—choosing $k$ out of $d_{out} \times d_{in}$ elements—results in

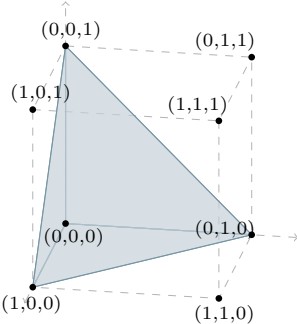 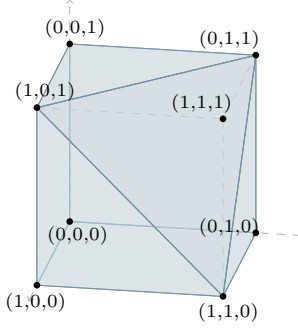

Figure 1: Visualization of $\mathcal{C}_k$ for $d_{\text{out}} = 3$, $d_{\text{in}} = 1$. **Left**: $k = 1$, **Right**: $k = 2$.

a search space that grows exponentially with the parameter count. Prior methods such as SparseGPT and Wanda therefore resort to greedy heuristics that ignore weight interactions to remain tractable[1].

In this work, we instead consider the convex relaxation of these combinatorial constraints: we approximate (MASK SELECTION) by optimizing over the convex hull of all masks, transforming the combinatorially hard problem into a tractable convex program

$$\min_M \|WX - (M \odot W)X\|_F^2, \quad \text{s.t. } M \in [0,1]^{d_{out} \times d_{in}}, \|M\|_1 \leq k \quad \text{(RELAXED MASK SEL.)}$$

where $M$ is now continuous with entries in $[0, 1]$, and the cardinality constraint is replaced by an $L_1$-norm budget, see Figure 1 for a visualization. The resulting convex program can be solved efficiently using the first-order Frank-Wolfe (FW) algorithm (Lacoste-Julien et al., 2013; Zeng & Figueiredo, 2014; Carderera et al., 2021; Braun et al., 2022). Notably, FW is projection-free and moves toward extreme points of the feasible set (i.e., binary masks) via a Linear Minimization Oracle (LMO), which is efficient to compute and naturally yields sparse updates.

Our method, which we term SparseFW, reduces the per-layer pruning error by up to 80% compared to state-of-the-art methods such as Wanda (Sun et al., 2023), and outperforms them on benchmark GPT architectures such as Qwen 2.5, LLaMA 3, Yi 1.5, and Gemma 2, with consistent gains in final WikiText perplexity and zero-shot accuracy. SparseFW is efficient, requires little memory overhead, easily adapts to unstructured and semi-structured sparsity patterns, is simple to implement, and scales to large models. Furthermore, unlike competing methods, SparseFW comes with strong theoretical justification: we show that, combined with the convergence guarantees of FW, rounding the relaxed solution to integrality yields an approximate solution to the original combinatorial problem.

**Contributions.** We summarize our contributions as follows.

1. **SparseFW: A projection-free method for layerwise pruning.** We formulate the layerwise mask selection problem as a convex program over the convex hull of binary masks and propose to solve it with the Frank-Wolfe (FW) algorithm, which is projection-free and leverages an efficient LMO that naturally yields sparse updates. SparseFW is memory-efficient, simple to implement, scales to large models, and can be used to induce both unstructured and semi-structured sparsity patterns.

2. **Strong empirical performance at LLM scale.** SparseFW reduces the per-layer pruning error by up to 70% compared to state-of-the-art methods such as Wanda, and delivers consistent gains in final WikiText perplexity and zero-shot accuracy across modern GPT architectures (e.g., Qwen 2.5, LLaMA 3, Yi 1.5, Gemma 2).

3. **Theoretical guarantees.** We provide approximation guarantees that connect the relaxed solution returned by FW after rounding to integrality to an approximate solution of the original combinatorial mask selection problem.

Our work demonstrates that classical constrained optimization techniques are not only feasible for pruning LLMs but can drastically improve upon state-of-the-art performance.

---

[1]We discuss these methods in detail in Section 2.

**Related work.** *Pruning after training* (Hoefler et al., 2021) is among the most popular approaches to reduce the resource demands of neural networks during inference. *Magnitude pruning* (Janowsky, 1989; Han et al., 2015) is the de facto default pruning criterion for convolutional architectures, and has been shown to yield pruned models that perform competitively, despite its simplicity (Gale et al., 2019; Zimmer et al., 2023). Various other criteria exist to decide which weights to consider unimportant (cf. LeCun et al., 1989; Hassibi & Stork, 1993; Molchanov et al., 2016; Yeom et al., 2019). With the rise of LLMs, magnitude pruning is being replaced by criteria that account for the peculiarities of LLMs (in particular, large activation outliers, cf. e.g. Dettmers et al., 2022; Yin et al., 2023) and that aim to avoid requiring retraining (Kwon et al., 2022; Frantar & Alistarh, 2023; Sun et al., 2023), which is generally considered computationally prohibitive for large models. Most importantly for our work, SparseGPT (Frantar & Alistarh, 2023), Wanda (Sun et al., 2023), and RIA (Zhang et al., 2024) address the mask selection problem (MASK SELECTION) using a greedy pruning approach, where the selection of weights to prune is performed iteratively. Our approach, on the other hand, relaxes the combinatorial constraint and takes weight interactions into account.

*Frank-Wolfe (FW)* or *conditional gradient* algorithms (Frank et al., 1956; Levitin & Polyak, 1966) are widely used in Machine Learning for handling complex structural requirements efficiently (Lacoste-Julien et al., 2013; Zeng & Figueiredo, 2014; Frandi et al., 2015; Jaggi, 2013; Négiar et al., 2020), with numerous theoretical works (Lacoste-Julien, 2016; Hazan & Luo, 2016; Reddi et al., 2016) and accelerated variants (Hazan & Luo, 2016; Yurtsever et al., 2019; Shen et al., 2019; Combettes et al., 2020; Mokhtari et al., 2018; Chen et al., 2018) appearing in the literature. For a comprehensive review, see Braun et al. (2022). Recently, FW has been applied in the context of neural networks (Ravi et al., 2018; Xie et al., 2019; Berrada et al., 2018; Tsiligkaridis & Roberts, 2020), for training neural networks at scale (Pokutta et al., 2020; Pethick et al., 2025), and Miao et al. (2022) as well as Zimmer et al. (2025) use FW-variants for inducing sparsity throughout pretraining.

## 2 METHODOLOGY

We begin by discussing the preliminaries and demonstrating that three state-of-the-art LLM pruning methods, namely SparseGPT, Wanda, and RIA, address the mask selection problem (MASK SELECTION) using a greedy pruning approach. We then introduce the FW algorithm and our proposed method, SparseFW. Throughout this section, we use lowercase letters for scalars and vectors and uppercase letters for matrices ($W$, $X$, $M$). Matrix entries are denoted $W_{ij}$ for the element in row $i$, column $j$. Rows of matrices are denoted with lowercase subscripts: $w_i$ represents the $i$-th row of matrix $W$. We use slicing notation, e.g., $X_{j,:}$ denotes the $j$-th row of matrix $X$.

### 2.1 PRELIMINARIES AND GREEDY METHODS

Before discussing SparseGPT, Wanda, and RIA in detail, we first note that the objective in Equation (MASK SELECTION) decomposes into a sum of $d_{out}$ row-wise quadratic functions

$$\|WX - (M \odot W)X\|_F^2 = \sum_{i=1}^{d_{\text{out}}} \|(w_i - m_i \odot w_i)X\|_2^2, \tag{1}$$

with $w_i \in \mathbb{R}^{d_{in}}$ and $m_i \in \{0,1\}^{d_{in}}$ denoting the $i$-th row of $W$ and $M$, respectively. Under unstructured sparsity, the constraint in (MASK SELECTION) couples the rows, making the problem non-separable. In contrast, semi-structured patterns such as $n{:}m$ (prune $M - N$ per block of $M$ weights) enforce equal per-row sparsity levels and hence fully decouple the rows. For simplicity, we will mainly discuss the row-wise formulation of Equation (1) and drop the index $i$. We now analyze how SparseGPT, Wanda, and RIA tackle the mask selection problem (MASK SELECTION) through greedy pruning—removing one weight at a time. These methods are optimal for their single-weight pruning objective, effectively bypassing weight interactions to simplify the problem.

*SparseGPT* (Frantar & Alistarh, 2023) is arguably the most popular approach and is largely based on preceding work (Frantar et al., 2022) of the authors. In practice, it prunes small blocks of weights at a time to ensure scalability to large models, instead of single weights in isolation as suggested by the theory; we briefly describe the underlying approach based on single-weight pruning. Instead of focusing solely on mask selection, SparseGPT approximates the problem of finding a sparse replacement $\hat{w}$ for the weight vector $w$, thus combining the problems of mask selection and

reconstruction of remaining weights by solving

$$\min_{\hat{w}} \| w^\top X - \hat{w}^\top X \|_F^2, \quad \text{s.t. } \|\hat{w}\|_0 \leq k. \tag{2}$$

Since solving this problem exactly is intractable, SparseGPT follows a greedy procedure to approximately solve it: at each step it finds the optimal *single* weight to prune and the corresponding optimal remaining weights, i.e., it solves

$$\min_{\hat{w}, q \in [d_{\text{in}}] \text{ s.t. } e_q^\top \hat{w} = 0} \| (\hat{w} - w)^\top X \|_2^2. \tag{3}$$

The greedy-best weight index $q$ and the optimal weight reconstruction are then given by

$$w^* = w - \frac{w_q}{[(XX^\top)^{-1}]_{qq}} (XX^\top)^{-1} e_q, \text{ where } q \in \arg\min_{q \in [d_{\text{in}}]} \frac{w_q^2}{((XX^\top)^{-1})_{qq}}.$$

*Wanda* (Sun et al., 2023) computes a saliency score $S_{i,j} := |W_{i,j}| \|X_{j,:}\|_2$ for each weight and then prunes the weights with the smallest saliencies. The authors motivate their approach by the observation that in LLMs, some weights with small magnitudes correspond to large-magnitude features (cf. e.g. Dettmers et al., 2022) and that their removal can lead to significant performance drops, despite their small magnitude. Wanda hence multiplies magnitude saliencies by the corresponding input activation norm to avoid pruning such small-but-important weights.

We argue that Wanda can be seen as a greedy approximation to (MASK SELECTION) and focus on a single row $w$ for simplicity. Again, we write the optimization problem for pruning one variable, but now without modifying the remaining weights:

$$\min_{\hat{w} = (1 - e_q) \odot w, \, q \in [d_{\text{in}}]} \left\{ \| (\hat{w} - w)^\top X \|_2^2 \right\} \tag{4}$$

Plugging the constraints into the objective function directly yields

$$\min_{q \in [d_{\text{in}}]} \left\{ \| ((1 - e_q) \odot w) - w)^\top X \|_2^2 \right\} = \min_{q \in [d_{\text{in}}]} \left\{ w_q^2 (XX^\top)_{qq} \right\} \tag{5}$$

Now note that $w_q^2 (XX^\top)_{qq} = w_q^2 \|X_{q,:}\|_2^2$. Minimizing the latter over $q$ is equivalent to minimizing $|w_q| \|X_{q,:}\|_2$, which is exactly the saliency score of Wanda.

While it might seem that this procedure differs from Wanda, as Wanda computes saliency scores once for all weights and not iteratively, the approaches are identical since the saliency scores do not change after pruning a weight. Wanda further enforces row-wise sparsity rather than unstructured sparsity, pruning a fixed number of weights per row. This has been found beneficial for LLMs (Sun et al., 2023); the same does not hold for other transformer-like models.

*RIA* (Zhang et al., 2024) builds upon Wanda and uses the following saliency score:

$$S_{ij}^{\text{RIA}} := |W_{ij}| \left( \frac{1}{\sum_{k=1}^{d_{\text{in}}} |W_{ik}|} + \frac{1}{\sum_{k=1}^{d_{\text{out}}} |W_{kj}|} \right) \|X_{j,:}\|_2. \tag{6}$$

We employ full-matrix notation since RIA fundamentally depends on the matrix structure for its row- and column-wise renormalization. Letting $W'$ denote the rescaled weight matrix with entries

$$W'_{ij} := W_{ij} \left( \frac{1}{\sum_{k=1}^{d_{\text{in}}} |W_{ik}|} + \frac{1}{\sum_{k=1}^{d_{\text{out}}} |W_{kj}|} \right).$$

Applying Wanda on $W'$ to prune the weights with the smallest saliency scores yields

$$|W'_{ij}| \|X_{j,:}\|_2 =: S_{ij}^{\text{RIA}}, \tag{7}$$

which is exactly the saliency score of RIA. The RIA criterion can be interpreted as using the same greedy pruning algorithm as Wanda, but applied to a rescaled weight matrix.

## 2.2 Solving the convex relaxation with Frank-Wolfe

We present an alternative approach to the greedy approximations discussed in the previous section, which is based on relaxing the combinatorial constraints to obtain a convex optimization problem, instead of trying to make the problem tractable by making the pruning decision on a per-weight basis. We solve the convex problem using the FW algorithm, which we introduce in the following.

**The Frank-Wolfe Algorithm.** When minimizing some objective function $\mathcal{L}$ over a set of constraints $\mathcal{C}$, a classical approach is Projected Gradient Descent (PGD) which iteratively performs a gradient step and then projects the result back to the constraint set to ensure feasibility of the iterates. However, depending on $\mathcal{C}$, this projection step may not admit an analytic solution and can be computationally expensive (Jaggi, 2013; Combettes & Pokutta, 2021). The FW algorithm is an alternative which is projection-free and often yields solutions with desirable structure. Instead of moving along the gradient direction and then requiring a projection step, FW moves towards the boundary point of the feasible region that is best aligned with the descent direction. Specifically, in each iteration $t$ and at iterate $M_t$, FW calls a Linear Minimization Oracle (LMO) on the gradient $\nabla \mathcal{L}(M_t)$ of $\mathcal{L}$ at $M_t$ to solve

$$V_t = \arg\min_{V \in \mathcal{C}} \langle V, \nabla \mathcal{L}(M_t) \rangle, \tag{8}$$

which is then used to update the parameters using the convex combination

$$M_{t+1} \leftarrow (1 - \eta_t) M_t + \eta_t V_t, \tag{9}$$

where $\eta_t \in [0, 1]$ is the step size. Throughout this work, we stick to the learning rate schedule given by $\eta_t = \frac{2}{t+2}$. If now $M_0 \in \mathcal{C}$, then the convex update rule ensures feasibility of all iterates. In practice, solving Equation (8) is often much cheaper than performing a projection step. If $\mathcal{C}$ is further given by the convex hull of a set of points, e.g., the vertices of a polytope, then the solution to Equation (8) is attained at one of these points. In each iteration, FW moves towards the vertices.

**Relaxing the combinatorial constraints.** The FW algorithm can only be applied to convex constraint sets, which is not the case for (MASK SELECTION). We make the problem tractable by relaxing the combinatorial constraints to their convex hull, i.e.,

$$\mathcal{C}_k = \left\{ M \in [0, 1]^{d_{\text{out}} \times d_{\text{in}}} : \|M\|_1 \leq k \right\}. \tag{10}$$

Given that the objective function of (MASK SELECTION) is a convex quadratic, this relaxation transforms the combinatorial mask selection problem into a convex optimization problem, which can be solved efficiently using the FW algorithm. We restate the reformulation of (RELAXED MASK SEL.) for completeness:

$$\min_{M \in \mathcal{C}_k} \|WX - (M \odot W)X\|_F^2. \tag{11}$$

This relaxation has the advantage that, unlike the previously discussed greedy approaches, it fully accounts for interactions between weights. However, the solution to the relaxed problem (RELAXED MASK SEL.) is not guaranteed to be feasible for the original problem (MASK SELECTION); in Section 4, we show that rounding the relaxed solution to integrality yields an approximate solution to the original problem.

**The sparse Linear Minimization Oracle.** We next discuss how to compute the LMO for the feasible set $\mathcal{C}_k$. Note that $\mathcal{C}_k$ is a polytope and can be described as the convex hull of its vertices, which are exactly the binary masks with at most $k$ ones. At any vertex, all coordinates lie on box bounds 0 or 1, and the coupling constraint $\sum_{i,j} M_{ij} \leq k$ is either inactive (fewer than $k$ ones) or tight (exactly $k$ ones); see Figure 1. Minimizing a linear function over $\mathcal{C}_k$ therefore consists of selecting up to $k$ entries with the most negative coefficients and setting them to one, leaving the rest at zero. Letting $\nabla \mathcal{L}(M_t) \in \mathbb{R}^{d_{\text{out}} \times d_{\text{in}}}$ denote the gradient of the objective at iterate $M_t$, the LMO solution at step $t$ is hence given by

$$[\text{LMO}\,(\nabla \mathcal{L}(M_t))]_{ij} = \begin{cases} 1 & \text{if } (i,j) \in \texttt{Top-k}\,(-\nabla \mathcal{L}(M_t)),\, [\nabla \mathcal{L}(M_t)]_{ij} < 0 \\ 0 & \text{otherwise} \end{cases}. \tag{12}$$

where $\texttt{Top-k}(\nabla \mathcal{L}(M_t))$ denotes the set of indices corresponding to the $k$ entries of $\nabla \mathcal{L}(M_t)$ with the smallest values. The LMO for $\mathcal{C}_k$ can be computed efficiently and naturally produces sparse updates: at most $k$ out of $d_{\text{out}} \cdot d_{\text{in}}$ entries are nonzero. While the above corresponds to unstructured sparsity, the LMO can be adapted to per-row sparsity and $n{:}m$ sparsity; see Appendix D.

## 2.3 THE SPARSEFW ALGORITHM

We present the full SparseFW algorithm in Algorithm 1. At a high level, for each layer we solve the relaxed optimization problem using the FW algorithm, starting from any binary mask that satisfies the sparsity constraints. After running for $T$ iterations, we threshold the learned mask—whose entries lie in $[0, 1]$—to obtain a binary mask that meets the original sparsity constraints. The objective function and the gradient with respect to $M_t$ are given by

$$\mathcal{L}(M_t) = \text{Tr}(W(1 - M_t)XX^\top(1 - M_t)^\top W^\top)$$

$$\nabla\mathcal{L}(M_t) = -2 \cdot W \odot (WXX^\top - (W \odot M_t)XX^\top).$$

Even for small calibration datasets, the activation matrix $X$ can be very large. For example, the largest matrix in a LLaMA-2-7B transformer block (`up_proj`) has $d_{in} = 4096$. With $N = 128$ samples and sequence length $L = 4096$, $X$ has dimensions $4096 \times 524{,}288$. Because both the objective and the gradient depend only on $G := XX^\top$ (which can be computed in batches), we precompute $G := XX^\top$ and $H := WG$ once to drastically reduce resource demands. Note that $G$ has dimensions $4096 \times 4096$, in contrast to the $4096 \times 524{,}288$ dimensions of $X$; this independence of the sequence length $L$ and number of samples $N$ is crucial for efficiency. With $G$ and $H$ precomputed, the gradient requires only two elementwise multiplications, a matrix–matrix multiplication, and a matrix addition:

$$\nabla\mathcal{L}(M_t) = -2 \cdot W \odot (H - (W \odot M_t)G).$$

In practice, we have to navigate a caveat that we did not detail in Algorithm 1 for the sake of simplicity, exact details are in the appendix. Throughout the experiments, we noticed that while FW often substantially reduces pruning error relative to baselines like Wanda, it can still produce worse final perplexity, likely due to a mismatch between local and global objectives. Constraining Sparse Frank-Wolfe (SparseFW) by fixing a fraction of very high-saliency weights (e.g., those with highest Wanda scores) as unprunable consistently improves performance. This suggests that Wanda reliably identifies weights that should be preserved, even if a more thorough local optimization would prune them. We therefore fix these weights and apply FW to the remaining ones, optimizing over a smaller search space. We ablate the impact of this ratio in Table 2 in the appendix: Surprisingly, we observe the best consistent improvements when setting $\alpha = 0.9$, i.e., fixing 90% of the highest saliency weights and optimizing only over the remaining 10%. Even small $\alpha$ values (e.g., $\alpha = 0.1$) can yield significant perplexity improvements. On the other hand, setting $\alpha = 0.0$ (full FW without any fixed weights) consistently yields worse results than the baselines.

---

**Algorithm 1** SparseFW

---

**Require:** Weight matrix $W$, input $X$, no. of nonzero entries $k$, iterations $T$, warm-start mask $M_0$

---

1: $G = XX^\top$, $H = WG$                           ▷ Precompute buffers
2: **for** $t = 0$ to $T - 1$ **do**
3:     $\nabla\mathcal{L}(M_t) = -2 \cdot W \odot (H - (W \odot M_t)G)$              ▷ Compute gradient
4:     $V_t = \text{LMO}(\nabla\mathcal{L}(M_t), \mathcal{C}_k)$                          ▷ Compute LMO
5:     $\eta_t \leftarrow \frac{2}{t+2}$
6:     $M_{t+1} \leftarrow (1 - \eta_t)M_t + \eta_t V_t$                       ▷ FW Update
7: $[M]_{ij} \leftarrow \begin{cases} 1 & \text{if } (i,j) \in \text{Top-k}(M_T) \\ 0 & \text{otherwise} \end{cases}$           ▷ Threshold
8: **return** $M$

---

## 3 EXPERIMENTAL RESULTS

We present our experimental methodology; our code will be made publicly available to ensure reproducibility. Our focus is on language modeling and we utilize pretrained models from Hugging-Face (Wolf et al., 2020), including *LLaMA-3.1-8B* (Grattafiori et al., 2024), *Gemma-2-9B* (Riviere et al., 2024), *Yi-1.5-9B* (Young et al., 2025), *DeepSeek-7B-base* (Bi et al., 2024), and *Qwen2.5-7B* (Yang et al., 2025). For the calibration set, we randomly sample 2048-token sequences from the *C4* dataset (Raffel et al., 2020). For validation, we select 100 sequences from the validation split. We evaluate performance using perplexity on *WikiText* (Merity et al., 2016) and zero-shot accuracy on

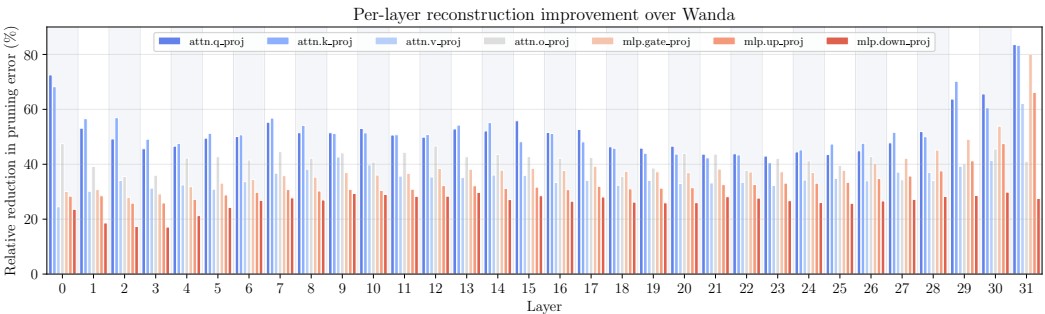

Figure 2: LLaMA-3.1-8B pruned to 60% unstructured sparsity with SparseFW using Wanda warmstart with 256 samples. This figure shows the relative reduction in pruning error (y-axis) for each matrix type (see legend for colors) for all layers of the model (x-axis) compared to the warmstart mask.

the EleutherAI evaluation set (Gao et al., 2023). Following Sun et al. (2023), we prune all linear layers with a uniform sparsity allocation across layers, while keeping the embedding and final linear head dense. SparseFW is compared with Wanda and RIA, as these methods also aim to find a better pruning mask by solving (MASK SELECTION); we hence do not compare directly to methods that involve a reconstruction step, such as SparseGPT (Frantar & Alistarh, 2023). We report results for both unstructured and semi-structured sparsity (Mishra et al., 2021).

**SparseFW outperforms state-of-the-art mask selection methods.** In Table 1, we compare SparseFW (warm-started from Wanda or RIA) to the respective baselines across five state-of-the-art GPTs and multiple sparsity regimes (50%, 60%, and 2:4). SparseFW generally performs on par with or better than the baselines in terms of perplexity; for zero-shot accuracy, SparseFW consistently outperforms competing methods. We generally observe much more consistent and bigger improvements in the higher sparsity regimes than for 50% sparsity.

**SparseFW successfully optimizes the matrix-wise pruning objective.** We observe consistent improvement in terms of the local pruning objective over both Wanda and RIA warmstarts. Figure 2 shows the per-layer reductions relative to a Wanda Warmstart, where we observe reductions of up to 80%. In general, we found the average relative reduction over the layers to range between 20% and 40% across the different models, sparsity regimes and warmstarts.

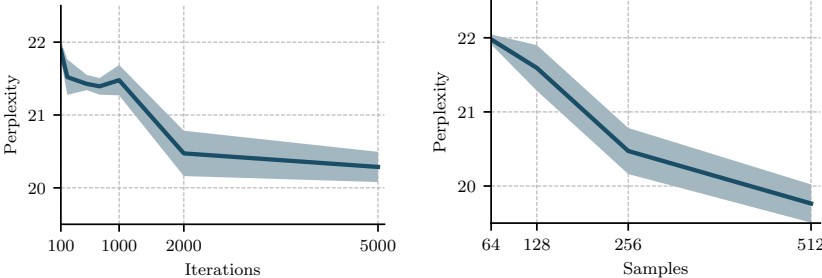

Figure 3: LLaMA-3.1-8B pruned to 2:4 sparsity using SparseFW. Left: Perplexity over the number of SparseFW iterations per layer with 256 samples. Right: Perplexity over the number of calibration samples with 2000 SparseFW iterations per layer. The solid curve represents the mean over multiple random seeds, the shaded regions represent the min-max range.

**Sample and iteration efficiency.** Figure 3 ablates the impact of the number of SparseFW iterations (left) and the number of calibration samples (right). Fixing the amount of samples at 256, perplexity decreases up to around 2000 iterations and then flattens. We therefore use 2000 iterations throughout. In contrast, at a fixed 2000 iterations, increasing the number of calibration samples from 64 to 512 brings substantial additional perplexity gains. This trend contrasts with Wanda, whose performance

Table 1: Perplexity ($\downarrow$, lower is better) and zero-shot accuracy ($\uparrow$, higher is better) comparison. We report SparseFW performance with Wanda and RIA warmstart for unstructured 50% and 60% sparsity and semi-structured 2:4 sparsity after 2000 iterations using 256 samples compared to the baseline warmstarts. We indicate the SparseFW warmstart method in parentheses. Best values are highlighted in bold. We omit standard deviations for legibility.

| Perplexity ($\downarrow$) | | GEMMA-2 | YI-1.5 | DEEPSEEK-7 | QWEN2.5 | | LLAMA-3 |
|---|---|---|---|---|---|---|---|
| Method | Sparsity | 9B | 9B | 7B | 7B | 14B | 8B |
| Wanda | | 11.19 | 6.58 | **7.79** | 8.45 | 7.11 | 10.09 |
| RIA | 50% | 11.19 | 6.71 | 7.90 | 8.54 | 7.01 | **9.88** |
| SparseFW (Wanda) | | **10.67** | 6.58 | 7.89 | 8.35 | 7.10 | 10.21 |
| SparseFW (RIA) | | 10.77 | **6.53** | 7.93 | **8.22** | **6.98** | 9.95 |
| Wanda | | 16.46 | 11.38 | **11.44** | 13.47 | 10.87 | 21.53 |
| RIA | 60% | 17.17 | 14.37 | 11.87 | 12.86 | 9.78 | 19.14 |
| SparseFW (Wanda) | | **14.83** | **10.56** | 11.99 | 12.44 | 10.28 | **17.97** |
| SparseFW (RIA) | | 15.07 | 10.67 | 12.41 | **11.66** | **9.65** | 18.16 |
| Wanda | | 17.41 | 11.58 | 11.76 | 14.40 | 11.37 | 24.82 |
| RIA | 2:4 | 16.78 | 11.27 | 12.04 | 13.46 | **10.98** | 23.7 |
| SparseFW (Wanda) | | **15.81** | 10.61 | **11.73** | 14.16 | 11.82 | **20.45** |
| SparseFW (RIA) | | 15.83 | **10.35** | 11.91 | **13.42** | 11.20 | 21.31 |
| **Accuracy in %** ($\uparrow$) | | GEMMA-2 | YI-1.5 | DEEPSEEK-7 | QWEN2.5 | | LLAMA-3 |
| Method | Sparsity | 9B | 9B | 7B | 7B | 14B | 8B |
| Wanda | | 68.44 | 61.04 | 56.67 | 63.72 | 67.94 | 58.78 |
| RIA | 50% | **68.71** | 61.22 | 55.76 | 64.03 | 67.83 | 58.94 |
| SparseFW (Wanda) | | 68.42 | 62.49 | **56.8** | 64.97 | **69.44** | **60.17** |
| SparseFW (RIA) | | 68.67 | **62.53** | 56.24 | **65.34** | 69.19 | 59.63 |
| Wanda | | 63.19 | 53.7 | 50.51 | 59.44 | 63.58 | 48.08 |
| RIA | 60% | 63.19 | 53.7 | 50.51 | 59.44 | 63.58 | 48.08 |
| SparseFW (Wanda) | | 64.46 | **54.90** | 50.56 | 61.13 | 65.59 | **51.92** |
| SparseFW (RIA) | | **65.35** | 55.41 | **50.65** | **61.52** | **65.80** | 52.15 |
| Wanda | | 63.75 | 52.92 | 50.65 | 59.11 | 63.39 | 47.13 |
| RIA | 2:4 | 63.83 | 52.41 | 51.08 | 58.48 | 63.85 | 47.77 |
| SparseFW (Wanda) | | 63.81 | **53.78** | **51.12** | 60.15 | 64.12 | 48.43 |
| SparseFW (RIA) | | **63.90** | 52.54 | 50.69 | 60.15 | **64.35** | **48.54** |

does not seem to increase significantly with additional calibration data: increasing the sample count from 64 to 512 leads to a perplexity decrease from 25.1 to only 24.6 for Wanda. Overall, SparseFW is clearly more compute-intensive than Wanda and RIA, but we argue that spending more resources once to improve the performance of pruned models is, given that deployed LLMs now serve millions of users and inference costs scale with the number of requests, worthwhile. That being said, the results of Figure 3 indicate clear benefits of increasing the number of samples while keeping the number of iterations fixed and relatively low. While more samples require slightly more compute to build the matrix $G = XX^\top$, the cost of a single FW iteration is independent of the sample count.

## 4 THEORETICAL RESULTS

In this section, we state a data-dependent error guarantee for the mask produced by SparseFW with respect to the original pruning objective (MASK SELECTION). This is a key benefit of SparseFW over greedy heuristics, which can yield suboptimal solutions even though the objective function is convex. We state our main result informally here, deferring full statements and proofs to the appendix.

**Lemma 1** (Informal). *After $T$ iterations of SparseFW, the resulting mask $M$ satisfies*

$$\mathcal{L}(M) - \mathcal{L}(M^*) \leq \lambda_{\max}(Q)\left(\frac{k}{T} + 2\left(k + \sqrt{2d_{in}d_{out}k}\right)\right)$$

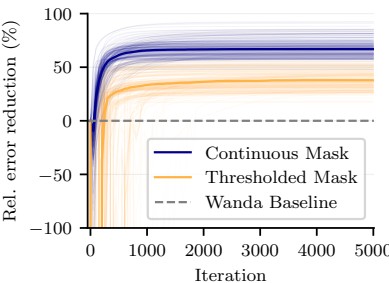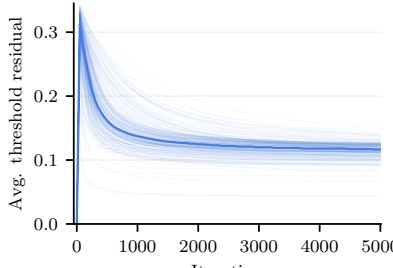

Figure 4: LLaMA-3.1-8B optimized towards 60% unstructured sparsity with SparseFW using 256 calibration samples. Lightly colored curves show the results individual matrices; the solid curve is their median. Left: Relative pruning error reduction versus FW iterations for continuous and thresholded masks. Right: Average threshold residual (mean $\ell_1$ distance between continuous and thresholded masks) versus iterations.

*where $M^*$ is an optimal mask for* (MASK SELECTION)*, $k$ is the maximum number of nonzeros in the mask, $Q$ represents the Hessian of the objective function and $\lambda_{\max}(Q)$ its largest eigenvalue.*

Note that $Q$ is not equal to $G = XX^\top$, the latter being the Hessian of the objective w.r.t. reconstruction of the weights, not w.r.t. the mask. The bound captures two sources of error: (i) the *optimization error* from solving the relaxed problem (RELAXED MASK SEL.), and (ii) the *thresholding error* from converting a relaxed solution to a binary mask (Line 7 in Algorithm 1).

*Optimization error.* After $T$ iterations of the FW algorithm, the resulting (continuous, not-yet-thresholded) mask $M_T$ satisfies

$$\mathcal{L}(M_T) - \mathcal{L}(\hat{M}) \leq k\lambda_{\max}(Q)/T,$$

where $\hat{M}$ is an optimal solution to the relaxed problem (RELAXED MASK SEL.). In other words, by increasing the number of iterations $T$, FW can guarantee an arbitrarily small optimization error.

*Thresholding error.* The error due to thresholding can be controlled by the curvature of the objective (captured by $\lambda_{\max}(Q)$) and the distance between the fractional iterate and its thresholded version, which in turn can be upper bounded in terms of $k$ and the dimension of the input space $d_{\mathrm{in}}d_{\mathrm{out}}$.

These insights explain the empirical behavior in Figure 4. The left panel reports the relative pruning error reduction (higher is better) versus FW iterations for the continuous and thresholded masks. After a short initial drop, due to the large stepsize, the continuous iterate improves consistently, as predicted by the FW convergence guarantee. In contrast, the thresholded mask first degrades as the thresholding error grows while the iterate moves through the interior of $\mathcal{C}_k$. This is reflected in the right panel, which shows the average threshold residual (the $\|\cdot\|_1$ distance between the continuous and thresholded masks): It first rises steeply, then decreases and eventually plateaus above zero. As long as the relaxed solution is not at a vertex, the thresholding error remains nonzero, so the thresholded curve does not fully catch up to the continuous one.

## 5 CONCLUSION

Solving the pruning mask selection problem for LLMs is a hard combinatorial problem. In this work, we relax the binary constraints to their convex hull and solve the resulting convex problem with the FW algorithm; we call this approach SparseFW, a simple and memory-efficient layerwise method that explicitly accounts for weight interactions and supports both unstructured and semi-structured sparsity. Across modern GPT architectures, SparseFW drastically reduces the per-layer reconstruction error and improves perplexity and zero-shot accuracy over state-of-the-art LLM pruning approaches. Our work demonstrates that classical constrained optimization is a scalable and effective alternative to greedy heuristics for LLM pruning.

However, our work is not without limitations. Although vanilla FW substantially reduces per-layer pruning error, this does not reliably yield lower perplexity. Without fixing part of the mask, it tends

to prune weights crucial for overall performance. SparseFW successfully mitigates this by preserving a fraction of high-saliency weights from the warmstart, but the local–global objective mismatch persists; inductive biases still appear necessary for improved perplexity.

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

# A USE OF LARGE LANGUAGE MODELS

Large language models were used to aid in writing (polishing text) as well as to help with the implementation of code components, including both the methods and the generation of plots. They also served as a tool for brainstorming research ideas and refining development approaches to address the challenges explored in this paper.

# B THE SPARSEFW ALGORITHM

We state the full SparseFW algorithm in Algorithm 2, which includes the details about how the fraction $\alpha$ of weights fixed to one is implemented. Before running FW, we compute the number of weights to keep based on saliency $k_{\text{keep}} = \lfloor k \cdot \alpha \rfloor$ and compute the mask of the weights to keep $\overline{M}$ by setting the $k_{\text{keep}}$ weights with the highest Wanda saliency scores $S$ to one and the remaining weights to zero. Then we apply FW to the remaining weights with the adjusted sparsity budget $k_{\text{new}} = k(1 - \alpha)$. Finally, we threshold the resulting mask $M_T$ by keeping its $k_{\text{new}}$ largest entries to obtain a binary mask $M^*$, and return $M^* + \overline{M}$, which preserves the salient weights and yields exactly $k$ nonzeros.

---

**Algorithm 2** The SparseFW algorithm

---

**Require:** Weight matrix $W$, input data $X$, nonzero entries $k$, maximum iterations $T$, warm-start saliency matrix $S$, fraction of weights to keep from saliency $\alpha$

---

1: $k_{\text{keep}} \leftarrow \lfloor k \cdot \alpha \rfloor$        ▷ Number of weights retained based on saliency
2: $k_{\text{new}} \leftarrow \lfloor k(1 - \alpha) \rfloor$        ▷ Remaining budget
3: $\overline{M}_{ij} \leftarrow 1$ for $(i, j) \in \texttt{Top-k}_{\text{keep}}(S)$, 0 otherwise        ▷ Fixed (preserved) mask
4: $G = XX^\top, H = WG$        ▷ Precompute caches
5: **for** $t = 0$ to $T - 1$ **do**
6:      $\nabla f(M_t) = -2 \cdot W \odot (H - (W \odot M_t)G)$        ▷ Compute gradient
7:      $V_t = \text{LMO}\big(\nabla f(M_t) \odot (1 - \overline{M}), \mathcal{C}_{k_{\text{new}}}\big)$        ▷ LMO on unfixed coordinates
8:      $\eta_t \leftarrow \frac{2}{t+2}$
9:      $M_{t+1} \leftarrow (1 - \eta_t)M_t + \eta_t V_t$        ▷ FW Update
10: $M_{ij}^* \leftarrow 1$ if $(i, j) \in \texttt{Top-k}_{\text{new}}(M_T)$ else 0        ▷ Threshold
11: **return** $M^* + \overline{M}$

---

## C  RATIO OF FIXED WEIGHTS ABLATION

Table 2 illustrates how the ratio $\alpha$ of fixed weights impacts SparseFW performance. Optimal results occur mostly at $\alpha = 0.9$, though even a small $\alpha$ (e.g., $\alpha = 0.1$) significantly enhance perplexity. Conversely, $\alpha = 0.0$ (full FW with no fixed weights) consistently underperforms compared to the baselines.

Table 2: Perplexity ($\downarrow$, lower is better) comparison on WikiText. We report SparseFW performance with after 2000 iterations using 256 samples with Wanda warmstart for unstructured 60% sparsity and semi-structured 2:4 sparsity for different ratios $\alpha$ of mask entries fixed to one (see Algorithm 2). Here, $\alpha = 1.0$ corresponds to the Wanda baseline, as no further mask entries can be optimized. Best values per row are highlighted in bold. The Wanda column provides a baseline for comparison.

| Model | Sparsity | $\alpha$-ratio of fixed weights | | | | | | 1.0 (Wanda) |
| | | 0.0 | 0.1 | 0.25 | 0.5 | 0.75 | 0.9 | |
| --- | --- | --- | --- | --- | --- | --- | --- | --- |
| Gemma-2-9B | 2:4 | 17.70 | 16.69 | 16.78 | 16.48 | 15.99 | **15.81** | 17.41 |
| Yi-1.5-9B | 2:4 | 12.26 | 11.50 | 11.49 | 11.25 | 10.83 | **10.61** | 11.58 |
| DeepSeek-7B | 2:4 | 13.25 | 12.77 | 13.13 | 12.99 | 12.32 | **11.73** | 11.76 |
| Qwen2.5-7B | 2:4 | 16.16 | 14.96 | 15.06 | 15.21 | 14.59 | **14.16** | 14.40 |
| Qwen2.5-14B | 2:4 | 13.70 | 12.62 | 13.34 | 12.99 | 12.79 | 11.82 | **11.37** |
| Llama-3.1-8B | 2:4 | 21.95 | 20.47 | **20.45** | 21.77 | 21.73 | 21.49 | 24.82 |
| Gemma-2-9B | 60% | 18.25 | 16.41 | 15.78 | 15.46 | 14.92 | **14.83** | 16.46 |
| Yi-1.5-9B | 60% | 11.19 | **10.56** | 10.66 | 10.81 | 11.06 | 11.31 | 11.38 |
| DeepSeek-7B | 60% | 12.49 | 11.99 | 12.06 | 12.19 | 12.20 | 12.21 | **11.44** |
| Qwen-7B | 60% | 14.28 | 13.13 | 13.12 | 12.73 | **12.44** | 12.54 | 13.47 |
| Qwen-14B | 60% | 11.59 | 10.52 | 10.48 | 10.61 | 10.29 | **10.28** | 10.87 |
| Llama-3.1-8B | 60% | 22.47 | 18.96 | **17.97** | 18.04 | 18.27 | 19.07 | 21.53 |

## D  LMOS FOR SEMI-STRUCTURED SPARSITY

Recall the definition of the constraint set $\mathcal{C}_k$ from Equation (10) for the unstructured sparsity case:

$$\mathcal{C}_k = \left\{ M \in [0,1]^{d_{\text{out}} \times d_{\text{in}}} : \|M\|_1 \leq k \right\}.$$

For the $n{:}m$ sparsity case, which corresponds to keeping at most $m$ nonzeros in every group of $n$ consecutive entries of each row, and assuming $d_{\text{in}}$ is divisible by $n$, we can write the constraint set as

$$\mathcal{C}_{n:m} = \left\{ M \in [0,1]^{d_{\text{out}} \times d_{\text{in}}} \ \Big|\ \sum_{j=qn+1}^{(q+1)n} M_{i,j} \leq m, \ \forall i, \ \forall q \in \{0, \ldots, d_{\text{in}}/n - 1\} \right\}.$$

Notice that this constraint set is simply the cartesian product of the constraint set for each block of $n$ consecutive entries of each row, which can be written as

$$\mathcal{C}' = \left\{ M' \in [0,1]^n : \|M'\|_1 \leq m \right\},$$

which is a special case of the polytope $\mathcal{C}_k$ when $d_{\text{out}} = n$, $d_{\text{in}} = 1$ and $k = m$. Since we know the LMO for $\mathcal{C}_k$ and the LMO problem is fully separable between the $\mathcal{C}'$ sets, we can simply apply the LMO for $\mathcal{C}_k$ to each set $\mathcal{C}'$ individually to obtain the LMO for $\mathcal{C}_{n:m}$.

## E  THEORETICAL GUARANTEE FOR SPARSEFW

For simplicity, we work in the row-wise formulation; the proof for the full-matrix case follows by the same arguments. Let us introduce the relevant notation and definitions for the row-wise formulation. We first fix $w \in \mathbb{R}_{\text{in}}^d$ (a row of $W$) and $X \in \mathbb{R}^{d_{\text{in}} \times B}$. For $m \in \mathcal{C}_k$ as defined in Equation (10), the objective function is

$$f(m) := \|w^\top X - (w \odot m)^\top X\|_2^2 = (1-m)^\top Q (1-m),$$

where $Q := \mathrm{Diag}(w)\,(XX^\top)\,\mathrm{Diag}(w) \succeq 0$. Let $\lambda_{\max}(Q)$ denote the top eigenvalue of $Q$. We denote the combinatorial constraint of the original problem Equation (MASK SELECTION) as

$$\mathcal{C}_{\mathrm{int}} := \Big\{ m \in \{0,1\}^{d_{\mathrm{in}}} \mid \sum_j m_j = k \Big\}.$$

Now we denote by $m^*$ the solution to the relaxed problem (RELAXED MASK SEL.) and by $m^{\mathrm{int}}$ the solution to the integral problem (MASK SELECTION).

**Lemma 2** (Formal statement of Lemma 1). *Let $m^\varepsilon \in \mathcal{C}_k$ satisfy $\sum_j m_j^\varepsilon = k$ and $f(m^\varepsilon) \le f(m^*)+\varepsilon$. Let $\widehat{m} := \mathbf{1}\{ j \in \mathrm{Top}\text{-}k(m^\varepsilon) \}$ be its top-k rounding. Then, with $r := d_{in} - k$,*

$$f(\widehat{m}) - f(m^{\mathrm{int}}) \le \varepsilon + 2\,\lambda_{\max}(Q)\Big( \min\{k,r\} + \sqrt{2r\,\min\{k,r\}}\Big). \tag{13}$$

*Note that for sparsity 50% or more, we have $2k \le d_{in}$ and hence $\min\{k,r\} = k$, it follows that*

$$f(\widehat{m}) - f(m^{\mathrm{int}}) \le \varepsilon + 2\lambda_{\max}(Q)(k + \sqrt{2d_{in} \cdot k}). \tag{14}$$

*Proof of Lemma 2.* Our goal is to bound $f(\widehat{m}) - f(m^{\mathrm{int}})$. To that end, first note that

$$f(m^\varepsilon) \le f(m^*) + \varepsilon \le f(m^{\mathrm{int}}) + \varepsilon, \tag{15}$$

where the first inequality follows by assumption on $m^\varepsilon$ and the second inequality follows since by the optimality of $m^*$ we have $f(m^*) \le f(m^{\mathrm{int}})$ (restricting to the $\mathcal{C}_{\mathrm{int}}$ can only make the objective worse). Therefore it suffices to bound $f(\widehat{m}) - f(m^\varepsilon)$.

Set $v := \widehat{m} - m^\varepsilon$ and $z^\varepsilon := \mathbf{1} - m^\varepsilon$. By construction, $\sum_j \widehat{m}_j = \sum_j m_j^\varepsilon = k$, hence $\mathbf{1}^\top v = 0$. Let

$$\tau := \sum_{j \notin \mathrm{Top}\text{-}k(m^\varepsilon)} m_j^\varepsilon = k - \sum_{j \in \mathrm{Top}\text{-}k(m^\varepsilon)} m_j^\varepsilon.$$

Then we have that

$$\begin{aligned}
f(\widehat{m}) - f(m^\varepsilon) &= (z^\varepsilon - v)^\top Q(z^\varepsilon - v) - (z^\varepsilon)^\top Q z^\varepsilon \\
&= v^\top Q v - 2 z^{\varepsilon\top} Q v \\
&\le \lambda_{\max}(Q)\,\|v\|_2^2 + 2\lambda_{\max}(Q)\,\|z^\varepsilon\|_2\,\|v\|_2 \\
&\le \lambda_{\max}(Q)(2\tau) + 2\lambda_{\max}(Q)\sqrt{r}\sqrt{2\tau},
\end{aligned}$$

where the equalities follow by defintion of $f$ and the first inequality follows by Cauchy-Schwarz. For the second inequality, consider that we have that $\|v\|_1 = 2\tau$ and $|v_j| \le 1$, hence $\|v\|_2^2 \le \|v\|_1 = 2\tau$. Further, we have that $\|z^\varepsilon\|_2^2 \le \|z^\varepsilon\|_1 = \sum_j(1 - m_j^\varepsilon) = d_{\mathrm{in}} - k = r$.

Lastly, we note that $\tau \le \min\{k,r\}$. This holds since we have that $\tau \le \sum_j m_j = k$ and

$$\tau = \sum_{j \notin \mathrm{Top}\text{-}k(m^\varepsilon)} m_j^\varepsilon \le \sum_{j \notin \mathrm{Top}\text{-}k(m^\varepsilon)} 1 \le d_{\mathrm{in}} - k$$

where the inequality follows since each $m_j \le 1$, and there are at most $d_{\mathrm{in}} - k$ terms in that sum. This concludes the proof for the Equation (13) and the proof of the Equation (14) follows by simple computations.

$\square$

