# OpenReview forum: "Don't Be Greedy, Just Relax! Pruning of LLMs via Frank-Wolfe"
_ICLR.cc/2026/Conference — Submitted to ICLR 2026_

### Official Review · Reviewer_JQdH · 2025-10-30

**Soundness:** 4
**Presentation:** 1
**Contribution:** 4
**Rating:** 0
**Confidence:** 5

**Summary:**

The paper is above 9 pages, therefore it should be rejected based on https://iclr.cc/Conferences/2026/AuthorGuide .

**Strengths:**

N/A

**Weaknesses:**

N/A

**Questions:**

N/A

---

### Official Review · Reviewer_b3v6 · 2025-10-31

**Soundness:** 1
**Presentation:** 1
**Contribution:** 1
**Rating:** 0
**Confidence:** 5

**Summary:**

The paper exceeds the 9-page limit for ICLR 2026. I believe it should be desk rejected.

**Strengths:**

N/A

**Weaknesses:**

N/A

**Questions:**

N/A

---

### Official Review · Reviewer_LJA4 · 2025-10-31

**Soundness:** 3
**Presentation:** 1
**Contribution:** 2
**Rating:** 0
**Confidence:** 5

**Summary:**

This paper proposed an LLM pruning method, named SparseFW, which solves mask selection via convex relaxation and the Frank-Wolfe algorithm. Unfortunately, the paper exceeds the 9-page length limit and is therefore subject to desk rejection, according to the Author Guideline: https://iclr.cc/Conferences/2026/AuthorGuide.

**Strengths:**

This paper proposed a novel LLM pruning method, SparseFW, that transforms combinatorial LLM pruning into tractable convex optimization via relaxation and the Frank-Wolfe algorithm, which supports multiple sparsity patterns.

**Weaknesses:**

The paper exceeds the 9-page length limit, as mentioned in the Author Guideline: https://iclr.cc/Conferences/2026/AuthorGuide.

**Questions:**

N/A

---

### Meta-Review · Area_Chair_8ak7 · 2026-01-04

**Summary:**

The paper exceeds the 9-page limit for ICLR 2026 and therefore should be desk rejected.

**Reviewer Concerns:**

The paper exceeds the 9-page limit for ICLR 2026; consequently, it was not reviewed by the reviewers.

**Reviewer Scores:**

N/A.

---

### Decision · Program_Chairs · 2026-01-26

Reject